# Stochastic Subnetwork Annealing: A Regularization Technique for Fine Tuning Subnetworks

## Abstract

Pruning methods have recently grown in popularity as an effective way to reduce the size and computational complexity of deep neural networks. Large numbers of parameters can be removed from trained models with little discernible loss in accuracy after a small number of continued training epochs. However, pruning too many parameters at once often causes an initial steep drop in accuracy which can undermine convergence quality. Iterative pruning approaches mitigate this by gradually removing a small number of parameters over multiple epochs. However, this can still lead to subnetworks that overfit local regions of the loss landscape. We introduce a novel and effective approach to tuning subnetworks through a regularization technique we call Stochastic Subnetwork Annealing. Instead of removing parameters in a discrete manner, we instead represent subnetworks with stochastic masks where each parameter has a probabilistic chance of being included or excluded on any given forward pass. We anneal these probabilities over time such that subnetwork structure slowly evolves as mask values become more deterministic, allowing for a smoother and more robust optimization of subnetworks at high levels of sparsity.

## 1 Introduction

Deep neural networks have seen a steady increase in size as large amounts of compute and high quality data have become more accessible. Models with billions of parameters are becoming commonplace and demonstrating incredible performance across a wide range of difficult machine learning tasks. However, these models also bring important challenges related to computational needs, storage cost, and training efficiency. The resource-intensive nature of these large networks have spurred a growing interest in techniques that can reduce the size and computational complexity associated with training and deploying these models.

One of the most popular methods used to compress large networks is weight pruning. It's long been known that you can remove a significant number of parameters from these trained models, and after a small number of epochs of continued training, they can maintain or even exceed the performance of the full size network (Blalock et al., 2020; Frankle & Carbin, 2019). While saliency metrics are often used to prune the most unnecessary weights, even random sampling has been show to produce accurate models at moderate levels of sparsity (Liu et al., 2022b).

Several researchers have investigated ways tune these subnetworks more efficiently and with better accuracy. One of the most effective techniques used in state of the art pruning methods involves iterative pruning/tuning cycles (Blalock et al., 2020). The key insight being that pruning too many parameters at once can lead to drastic performance collapse as subnetworks get stuck in lower performing regions of the optimization landscape. Pruning a small number of parameters over several epochs allows the model to better adapt to the changing network structure. However, these iterative methods can still result in subnetworks that are prone to overfitting local optimization regions.

We introduce a novel regularization approach for fine tuning these subnetworks by leveraging stochasticity for the network structure. Rather than using fixed subnetworks and discrete pruning operations, we instead represent subnetworks with probability matrices that determine how likely it is that a parameter is retained on any given forward pass. The probability matrices are then adjusted

during the tuning phase through the use of dynamic annealing schedules which allows for a subnetwork to be slowly revealed over several epochs. The probabilistic inclusion of extra parameters early in the tuning process allows for gradient information to bleed through into the target subnetwork, encouraging robust adaptation and avoiding the drastic performance collapse observed with one-shot pruning methods.

We conduct a large scale ablation study to explore the efficacy and dynamics of several hyperparameters and their effects on convergence, including initial stochasticity, number of annealing epochs, amount of sparsity, constant/phasic learning rate schedules, and random/saliency based parameter selection strategies. Our experiments demonstrate significant improvements over both one-shot and iterative pruning methods across several tuning configurations with especially large improvements in highly sparse subnetworks (95-98%).

We additionally explore how Stochastic Subnetwork Annealing can be leveraged in a benchmark low-cost ensemble learning method that generates diverse child networks through random pruning of a trained parent network. Implementing our technique to tune the child networks results in better ensemble generalization on benchmark image classification tasks, illustrating a new Pareto Frontier for the computational efficiency/accuracy boundary.

## 2  BACKGROUND

Subnetworks are represented with binary bit mask matrices with the same dimensions of the weight matrices for each layer in the parent network. Consider a weight matrix $W \in \mathbb{R}^{m \times n}$ representing the weights of a particular layer in a neural network. We introduce a matrix $M \in \{0, 1\}^{m \times n}$ with the same dimensions as W. The elements of M are binary values, where $M_{ij} = 1$ if the corresponding weight $W_{ij}$ is retained, and $M_{ij} = 0$ if the weight is masked. The mask is generated with an arbitrary discrete stochastic process $\phi$. The subnetwork weights $\hat{W}$ can then be computed as the element-wise (Hadamard) product of W and M.

$$M \sim \phi^{m \times n} \tag{1}$$

$$\hat{W} = W \circ M \tag{2}$$

The topology of the subnetwork can be further described by the granularity in which parameters are masked. This granularity refers to the unstructured or structured distribution of masked weights, where unstructured methods refer to weight-level or connection-level masking and structured methods refer to neuron-level, channel-level, or layer-level masking. Removing entire rows, columns, or blocks from a layer's weight matrix can be effective at reducing computational complexity as the reduced size of the weight matrix can be leveraged for hardware optimizations. This is more difficult to achieve with the sparse matrices resulting from unstructured masking. However, unstructured masking tends to result in networks with better generalization as the number of masked parameters increases (Blalock et al., 2020).

$$\hat{W}_{uns} = \begin{bmatrix} w_{11} & w_{12} & 0 & w_{14} \\ 0 & w_{22} & 0 & 0 \\ w_{31} & 0 & 0 & w_{34} \\ 0 & 0 & w_{43} & w_{44} \end{bmatrix} \tag{3}$$

$$\hat{W}_{str} = \begin{bmatrix} w_{11} & w_{12} & w_{13} & w_{14} \\ 0 & 0 & 0 & 0 \\ w_{31} & w_{32} & w_{33} & w_{34} \\ 0 & 0 & 0 & 0 \end{bmatrix} \tag{4}$$

It's important to also consider the distribution of masked weights throughout non-homogenous networks. It's common for layer configurations in deep neural networks to vary significantly in size and shape. Severe pruning of small layers may result in bottlenecks that restrict gradient flow. The distribution of masked weights can be controlled with global and local masking methods. Global methods are applied uniformly across the entire network while local methods are applied independently within each layer or sub-region. Global methods tend to result in higher compression rates as more parameters from the larger layers are removed whereas local methods offer more fine-grained control and reduced variance (Blalock et al., 2020).

Continued training of the subnetwork is crucial in order to recover the lost accuracy from pruning. Only a small number of epochs are generally needed as subnetworks inherited from trained networks converge quickly. The standard practice for fine-tuning subnetworks involves training with a small learning rate consistent with the final phase of the parent network's training (Li et al., 2017; Liu et al., 2019; Han et al., 2015).

Iterative pruning has been shown to be highly effective at improving accuracy of subnetworks compared to one-shot pruning (Li et al., 2017; Frankle & Carbin, 2019; Han et al., 2015; Gale et al., 2019). Instead of pruning all of the weights at once, instead an iterative cycle is implemented where a small number of weights are pruned, the network is tuned, and this repeats until a target sparsity level is reached. Iterative pruning results in a less destructive effect on network performance which can allow for greater levels of sparsity at improved accuracy.

The Lottery Ticket Hypothesis introduced the concept of weight rewinding, where both the network weights and the learning rate is rewound to a previous state $t$ epochs ago. Training continues from this previous state but with the new subnetwork structure fixed (Frankle & Carbin, 2019). However, rewinding the weights have been shown to be less effective than rewinding only the learning rate (Renda et al., 2020). The efficacy of learning rate rewinding has been tangentially shown in many optimization papers where a decaying learning rate schedule has been shown to improve training efficiency and convergence quality in deep networks (You et al., 2019).

Cyclic learning rate schedules have been used to good effect in several low-cost ensemble methods, where large learning rates can help to encourage more diversity by moving further distances in parameter space before using small learning rates to converge to local optima (Huang et al., 2017; Garipov et al., 2018; Whitaker & Whitley, 2022). We experiment with a learning rate schedule called the one-cycle policy, which consists of a warm-up phase that anneals from a small learning rate to a large learning rate for the first 10% of training, followed by a cosine annealed cool-down to 0. This schedule has been shown to lead to a phenomenon called super-convergence, where network training is greatly accelerated on some datasets (Smith & Topin, 2018).

The value for the learning rate $\eta$ at iteration $t$, where $\eta_{init}$ is the initial learning rate value, $\eta_{max}$ is the maximum value, $\eta_{min}$ is the minimum value and $T$ is the total number of iterations can be described as:

$$\eta_{warm}(t) = \eta_{init} + \frac{1}{2}(\eta_{max} - \eta_{init})\left(1 - cos\left(\frac{\pi t}{T}\right)\right) \tag{5}$$

$$\eta_{cool}(t) = \eta_{min} + \frac{1}{2}(\eta_{max} - \eta_{min})\left(1 + cos\left(\frac{\pi t}{T}\right)\right) \tag{6}$$

## 3 STOCHASTIC SUBNETWORK ANNEALING

In pruning literature, sparse network structures are generally static and represented with binary bit masks. We propose a model of representing neural subnetworks with probabilistic masks, where each parameter is assigned a score that determines how likely it is that the parameter will be retained on any given forward pass. This introduces stochasticity into the subnetwork sampling process, which can act as a form of implicit regularization analagous to a reverse dropout, where parameters that would have been pruned have a chance to activate. This technique encourages exploration of a larger space of subnetworks during the fine-tuning phase, preventing it from becoming overly reliant on a single fixed topological configuration, resulting in more robust and generalized subnetworks.

Consider a weight matrix $W \in \mathbb{R}^{m \times n}$ representing the weights of a particular layer in a neural network. We introduce a probability matrix $P \in \mathbb{R}^{m \times n}$ containing scores that represent the probability that a parameter $W_{ij}$ will be will be masked on any given forward pass. The subnetwork mask $M \in \{0, 1\}^{m \times n}$ is determined with a Bernoulli realization of the probability matrix $P$.

We then anneal these probability values over some number of epochs such that subnetworks are slowly "revealed" throughout fine-tuning. This is done by introducing an annealing schedule, where the probability values for each parameter slowly move towards 0 or 1 depending on the target subnetwork sparsity. At the beginning of the training process, a high level of stochasticity is desirable as it encourages exploration of the weight space which may help to prevent overfitting and search for more effective subnetwork configurations. As training progresses, the stochasticity should gradu-

ally decrease to allow for stable convergence. The annealing function is arbitrary, with some popular examples being the linear, cosine, and exponential decay.

## 3.1 RANDOM ANNEALING

The probability matrix can be generated in a variety of ways. For example, a uniform distribution $P \sim U([0,1])^{m \times n}$ can be used to randomly assign probabilities to each parameter. All parameters with a value less than the sparsity target will then anneal towards 0 while parameters with a probability value greater than the sparsity target will anneal towards 1.

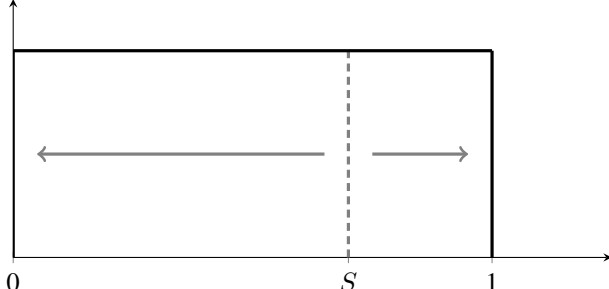

With this implementation, the mean activation of parameters at the beginning of tuning will be 50% regardless of the target subnetwork sparsity. Other distributions can offer more fine grained control over network structure. For example, assume that a binary matrix $X \in \{0,1\}^{m \times n}$ is randomly generated and used to index into a probability matrix $P \in \mathbb{R}^{m \times n}$. Using this index matrix $X$, we can sample from Gaussian distributions with different means and variances.

$$P = \begin{cases} P_{ij} \sim \mathcal{N}(\mu_1, \sigma_1^2), \text{ if } X_{ij} = 0 \\ P_{ij} \sim \mathcal{N}(\mu_2, \sigma_2^2), \text{ if } X_{ij} = 1 \end{cases}$$

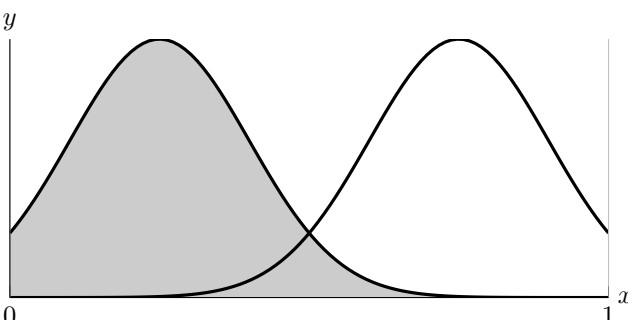

This approach to constructing multi-modal distributions can allow for many interesting formulations of stochastic subnetworks. Future work may find natural applications to multi-task learning, where certain groups of parameters can be strongly correlated with each other while allowing for overlap with other task specific subnetworks. This stochastic overlap may encourage shared portions of the network to learn generalized features, avoiding the problem in typical network-splitting approaches where task specific subnetworks become increasingly narrow and lead to degraded performance.

## 3.2 TEMPERATURE ANNEALING

Temperature scaling may be applied to binary matrices in order to allow for an even application of stochasticity, reducing some variance relative to random annealing. Assume that some binary matrix $X \in \{0,1\}^{m \times n}$ is generated to represent a target subnetwork. A temperature scaling constant $\tau$ is introduced such that the values in $X$ with a 1 are decayed by $\tau$ and the values with 0 are increased by $\tau$. The mask is then determined on every forward pass with an altered probability according to $\tau$.

Altering the initial value of $\tau$ allows for a much more controlled approach to noise injection during the early phases of tuning, which can be highly desirable when the number of total tuning epochs is limited.

A variation of temperature scaling, where tau is only applied to parameters that are not a part of the target subnetwork, will be shown to be a highly effective form of regularization analagous to a reverse dropout. That is, the subnetwork is always active, but parameters that would have been pruned away now have a chance to pop back in during tuning. This formulation is less destructive than full temperature scaling as the target subnetwork will be optimized on every training step. Allowing other parameters to become active allows for gradient information to contribute to optimization of the target subnetwork which can help to encourage avoidance of local minima.

$$P = \begin{bmatrix} 1 & 1 & 0+\tau & 1 \\ 0+\tau & 1 & 0+\tau & 0+\tau \\ 1 & 0+\tau & 0+\tau & 1 \\ 0+\tau & 0+\tau & 1 & 1 \end{bmatrix}$$

## 4 STOCHASTIC SUBNETWORK ENSEMBLES

Ensemble learning is a powerful technique for improving the generalization of machine learning systems (Hansen & Salamon, 1990; Krogh & Vedelsby, 1994). These algorithms train multiple models which are then evaluated independently on test data. The combination of several predictions allows for bias and variance reduction that results in reliable performance improvement. However, as datasets and neural networks have grown larger, traditional ensemble methods have become prohibitively expensive to implement. Recent research has shown that low-cost ensemble methods can achieve performance that rivals full size ensembles at a significantly reduced cost. Several of the most powerful low-cost methods do this by leveraging sparse subnetworks within large parent networks. (Whitaker & Whitley, 2022; von Oswald et al., 2022; Liu et al., 2022a; Havasi et al., 2021).

Prune and Tune Ensembling (PAT) is one such technique that demonstrates incredible efficiency for the training compute/accuracy tradeoff. These work by first training a single parent network for the majority of the training budget. Child networks are then spawned by cloning and dramatically pruning the parent using random or anti-random sampling strategies. Each of the child networks are then fine tuned with a cyclic learning rate schedule for a small number of epochs.

As child networks are all derived from an identical parent network, anti-random pruning and one-cycle tuning are used to encourage diversity and reduce correlation among ensemble members by ensuring that topological structures are distant and that parameters move far apart in optimization space before converging (Malaiya, 1995; Wu et al., 2008; Smith & Topin, 2018; Whitaker & Whitley, 2022).

Anti-random pruning creates mirrored pairs of child networks, such that whenever we randomly prune the parent to create a child, a sibling is created that inherits the opposite set of parameters. Consider a binary bit string $M = \{x_0, ..., x_n : x \in \{0, 1\}\}$, that is randomly generated with 50% sparsity where 1 represents parameters that are kept and 0 represents parameters that are pruned. The anti-random network then is created by reversing the polarity of all the bits in the mask $M$, such that:

$$\hat{\theta}_1 = \theta \circ M \quad \text{and} \quad \hat{\theta}_2 = \theta \circ (1 - M) \tag{7}$$

where $\hat{\theta}_i$ are the parameters of the child network, $\theta$ are the parameters of the parent network and $\circ$ denotes the Hadamard product.

Stochastic Subnetwork Annealing can be easily implemented to tune child networks within the context of this low-cost ensemble algorithm. Our implementation of random annealing can naturally extend to the ideas of anti-random pruning through anti-probability matrices. When a probability matrix is generated, a mirrored probability matrix can be generated such that the subnetworks anneal to opposite topological structures where $P' = 1 - P$.

| Random/Constant | 50% | 70% | 90% | 95% | 98% |
|---|---|---|---|---|---|
| One-Shot Baseline | $64.0 \pm 0.07$ | $63.4 \pm 0.13$ | $54.9 \pm 0.12$ | $48.4 \pm 0.24$ | $41.1 \pm 0.47$ |
| Iterative Pruning | $65.1 \pm 0.10$ | $63.4 \pm 0.05$ | $57.8 \pm 0.06$ | $49.8 \pm 0.22$ | $42.8 \pm 0.26$ |
| Random Annealing | $65.3 \pm 0.17$ | $63.6 \pm 0.15$ | $58.9 \pm 0.17$ | $54.5 \pm 0.16$ | $45.7 \pm 0.22$ |
| Temperature Annealing | $65.6 \pm 0.09$ | $64.3 \pm 0.12$ | $59.5 \pm 0.02$ | $55.4 \pm 0.19$ | $46.5 \pm 0.15$ |

| Random/One-Cycle | 50% | 70% | 90% | 95% | 98% |
|---|---|---|---|---|---|
| One-Shot Baseline | $70.1 \pm 0.07$ | $68.9 \pm 0.19$ | $61.8 \pm 0.07$ | $56.8 \pm 0.21$ | $49.3 \pm 0.27$ |
| Iterative Pruning | $70.1 \pm 0.03$ | $69.2 \pm 0.11$ | $63.9 \pm 0.05$ | $57.4 \pm 0.12$ | $49.8 \pm 0.51$ |
| Random Annealing | $70.2 \pm 0.09$ | $69.2 \pm 0.12$ | $64.5 \pm 0.07$ | $59.7 \pm 0.16$ | $52.7 \pm 0.30$ |
| Temperature Annealing | $70.6 \pm 0.06$ | $69.6 \pm 0.09$ | $64.9 \pm 0.08$ | $60.2 \pm 0.08$ | $53.3 \pm 0.29$ |

| Magnitude/Constant | 50% | 70% | 90% | 95% | 98% |
|---|---|---|---|---|---|
| One-Shot Baseline | $67.1 \pm 0.16$ | $67.3 \pm 0.37$ | $65.7 \pm 0.24$ | $63.3 \pm 0.14$ | $54.4 \pm 0.24$ |
| Iterative Pruning | $68.2 \pm 0.09$ | $67.9 \pm 0.04$ | $65.9 \pm 0.07$ | $63.3 \pm 0.02$ | $56.2 \pm 0.11$ |
| Temperature Annealing | $68.5 \pm 0.03$ | $68.3 \pm 0.03$ | $66.3 \pm 0.02$ | $63.4 \pm 0.11$ | $56.6 \pm 0.19$ |

| Magnitude/One-Cycle | 50% | 70% | 90% | 95% | 98% |
|---|---|---|---|---|---|
| One-Shot Baseline | $70.9 \pm 0.12$ | $70.1 \pm 0.26$ | $69.7 \pm 0.06$ | $67.5 \pm 0.16$ | $60.2 \pm 0.10$ |
| Iterative Pruning | $71.1 \pm 0.04$ | $71.1 \pm 0.06$ | $69.9 \pm 0.16$ | $67.6 \pm 0.08$ | $60.8 \pm 0.19$ |
| Temperature Annealing | $71.5 \pm 0.06$ | $71.4 \pm 0.02$ | $70.3 \pm 0.10$ | $67.8 \pm 0.02$ | $61.5 \pm 0.02$ |

Table 1: Comparison between various baseline approaches when used with random/magnitude pruning on CIFAR-100. We report the best accuracy for each method at various levels of target sparsity when tuned with both a constant learning rate and a one-cycle learning rate schedule. The parent network has a baseline accuracy of 71.5%. Our annealing strategies consistently outperform established one-shot and iterative pruning methods.

## 5  EXPERIMENTS

### 5.1  ABLATIONS

We begin with an exploration of several stochastic subnetwork annealing configurations with the goal of investigating how different hyperparameters impact the efficiency and quality of subnetwork convergence, compared to established one-shot and iterative pruning techniques.

We use the benchmark CIFAR-10/CIFAR-100 datasets to conduct our explorations (Krizhevsky, 2012). CIFAR consists of 60,000 small natural colored images that are 32x32 pixels in size. Each dataset is split into a training set containing 50,000 images and a test set containing 10,000 images. CIFAR-10 samples images from 10 different classes, or target labels, while CIFAR-100 samples from 100 different classes. Thus, CIFAR-10 contains 5,000 images for each class while CIFAR-100 is comparatively more difficult containing only 500 images for each class.

All ablations use the same ResNet-18 trained for 100 epochs using a standardized optimization configuration (He et al., 2016). We use PyTorch's Stochastic Gradient Descent optimizer with an initial learning rate of $0.1$ and Nesterov momentum of $0.9$ (Sutskever et al., 2013). After 50 epochs, the learning rate is decayed to $0.01$ and again to $0.001$ for the final 10 epochs. We use standard data augmentations including random crop, random horizontal flip, and mean standard normalization.

Pruning is done in a layerwise unstructured fashion, after which each subnetwork is tuned for an additional 20 epochs. We experiment with both a constant learning rate of $0.01$ and a one-cycle policy with a max learning rate of $0.1$. We explore the results for each configuration with different levels of final subnetwork sparsity $\rho \in [0.5, 0.7, 0.9, 0.95, 0.98]$. We additionally include results for subnetworks created through L1 unstructured pruning. In this case, parameters with the smallest magnitudes at each layer are pruned.

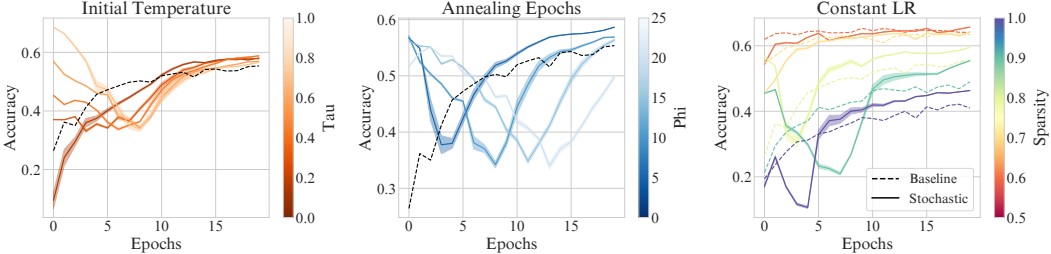

Figure 1: Ablations exploring how the initial temperature and the number of annealing epochs affect convergence behavior. The rightmost graph displays the best performing annealing configuration for each sparsity and plots it against the one-shot pruning baseline. Stochastic annealing outperformed the other methods for each sparsity, with more drastic improvements appearing in extremely sparse networks.

The one-shot baseline prunes the parent network to the target sparsity before we start tuning. Iterative pruning calculates the number of parameters to remove at the beginning of each epoch over some number of pruning epochs $\varphi \in [5, 10, 15, 20]$ such that the final target sparsity is hit. Random Annealing uses a probability matrix that is generated according to a random uniform distribution, where each parameter is assigned a value between 0 and 1. Parameters with a value less than the target sparsity value are linearly annealed to 0 and values greater than the target sparsity are linearly annealed to 1, over some number of annealing epochs $\varphi \in [5, 10, 15, 20]$. Temperature Annealing uses a randomly generated binary bitmask according to the target sparsity. All parameters with a value of 0 are modified to an initial temperature value of $\tau \in [0.2, 0.4, 0.6, 0.8, 1.0]$. Those values are then cosine annealed to 0 over some number of annealing epochs $\varphi \in [5, 10, 15, 20]$.

Table 1 includes the mean accuracies for the best configurations of each method on CIFAR-100. We see consistent improvement with both of our stochastic annealing methods over the baseline one-shot and iterative pruning techniques across all sparsities and with both a constant learning rate and a one-cycle policy. As networks become more sparse, the benefits from our annealing approaches become more significant, with a 6% and 4% improvement at 98% sparsity over the one-shot baseline with a constant and one-cycle rate schedule respectively. We also observed improved performance with magnitude pruning, however the differences were smaller at 2% and 1% improvement respectively.

Figure 1 includes an exploration of the initial temperature, the number of annealing epochs and the test trajectories for the best models tuned with a constant learning rate schedule. The hyperparameter with the most significant impact on all pruning methodologies was the number of epochs that were pruned or annealed over. We saw best results for all subnetwork sparsities with $e = 5$ or $e = 10$ annealing epochs. It's important for the final subnetwork topology to be established for a sufficient number of epochs to allow for optimal convergence behavior. This pattern holds for both constant and one-cycle learning rate schedules. The initial temperature has a smaller impact on performance than the number of annealing epochs. A small value of $\tau$ means that the target subnetwork is always active and other parameters have a small chance to turn on, while a high value of $\tau$ means that the target subnetwork is always active and other parameters have a high chance to turn on. We saw best results when $\tau$ was in the 0.4 to 0.6 range. This corresponds to a higher state of entropy regarding network structure which results in a stronger regularization effect for those initial training examples.

Figure 2 includes an exploration of random annealing vs temperature annealing, iterative pruning vs temperature annealing, and the test trajectories for the best models with a one-cycle learning rate schedule. Despite the additional variance associated with random annealing, we found that the performance was very good and nearly approached that of temperature annealing. Temperature annealing consistently outperformed iterative pruning. The graph displays results for annealing epochs $e \in [5, 10]$ and a target subnetwork sparsity of 0.9. While the early accuracy of temperature annealing appears worse, it quickly surpasses iterative pruning as the annealing phase ends. The One-Cycle schedule does reduce the accuracy gap between our method and the baselines due to the much improved generalization compared to the constant learning rate models. However, temperature annealing still consistently outperformed the baseline across all sparsities.

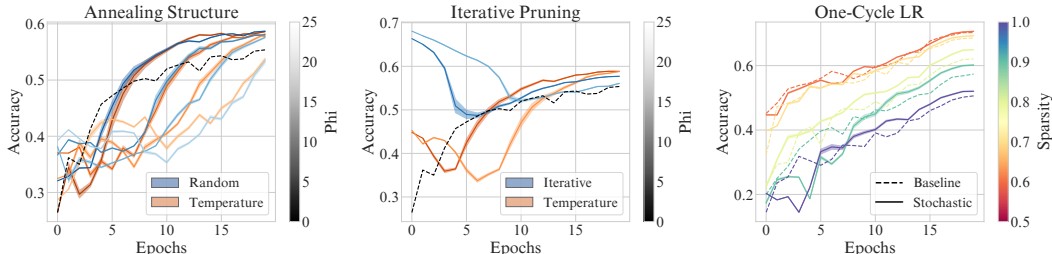

Figure 2: Ablations over the number of annealing epochs for random annealing vs temperature annealing and temperature annealing vs iterative pruning. The rightmost graph displays the best performing annealing configuration for each sparsity and plots it against the baseline. Stochastic annealing outperformed the other methods for each sparsity when used with a one-cycle policy.

## 5.2 STOCHASTIC SUBNETWORK ENSEMBLES

We aim to evaluate the efficacy of Stochastic Subnetwork Annealing in the context of the low-cost ensemble algorithm, Prune and Tune Ensembles. We include results for these ensembles on CIFAR-10, CIFAR-100, and corrupted versions of both in order to test robustness on out-of-distribution images. These corrupted datasets are generated by adding 20 different kinds of image corruptions (gaussian noise, snow, blur, pixelation, etc.) at five different levels of severity to the original test sets Hendrycks & Dietterich (2019); Nado et al. (2021). The total number of images in each of these additional sets is 1,000,000.

We take the training configuration, ensemble size and parameter settings directly from studies of three state-of-the-art benchmark low-cost ensemble methods: MotherNets (Wasay et al., 2018), Snapshot Ensembles (Huang et al., 2017), and Fast Geometric Ensembles (Garipov et al., 2018). We also compare our results with published results of several recent low-cost ensemble methods including: TreeNets (Lee et al., 2015), BatchEnsemble (Wen et al., 2020), FreeTickets (Liu et al., 2022a), and MIMO (Havasi et al., 2021).

All methods compared use WideResNet-28-10 and Stochastic Gradient Descent with Nesterov momentum and weight decay. The Sparse Subnetwork Ensemble size and training schedule is as used in previous comparisons (Wasay et al., 2018; Garipov et al., 2018). We use a batch size of 128 for training and use random crop, random horizontal flip, and mean standard normalization data augmentations for all approaches (Garipov et al., 2018; Havasi et al., 2021; Liu et al., 2021; Huang et al., 2017). The parent learning rate uses a step-wise decay schedule. An initial learning rate of $\eta_1 = 0.1$ is used for 50% of the training budget which decays linearly to $\eta_2 = 0.001$ at 90% of the training budget. The learning rate is kept constant at $\eta_2 = 0.001$ for the final 10% of training.

We train a single parent network for 140 epochs. Six children are then created by randomly pruning 50% of the connections in the parent network. Neural partitioning is implemented where pairs of children are generated with opposite sets of inherited parameters. Each child is tuned with a one-cycle learning rate for 10 epochs. The tuning schedule starts at $\eta_1 = 0.001$, increases to $\eta_2 = 0.1$ at 1 epoch and then decays to $\eta_3 = 1e - 7$ using cosine annealing for the final 9 epochs.

We implement Stochastic Annealing when tuning the child networks by using the procedure illustrated above for Temperature Annealing. We initialize the binary subnetwork masks with a target sparsity of $0.5$. We then modify the masks such that the 0 parameters are initialized with a temperature of $\tau = 0.5$. That value is decayed to $\tau = 0$ over 3 annealing epochs using a cosine decay.

We compare the results of our approach to a wide variety of competitive benchmarks. These benchmarks include both low-cost and full-size ensemble approaches. The details for these implementations are taken from baseline results reported in Whitaker & Whitley (2022); Havasi et al. (2021); Liu et al. (2022a), and is informed by each original implementation in Huang et al. (2017); Garipov et al. (2018); Lee et al. (2015); Wen et al. (2020); Havasi et al. (2021).

Table 2 reports the mean accuracy (Acc), negative log likelihood (NLL), and expected calibration error (ECE) over 3 runs on both CIFAR-10 and CIFAR-100 along with their corrupted variants. We report the total number of floating point operations (FLOPs) and epochs used for training each

| Methods (CIFAR-10/WRN-28-10) | Acc ↑ | NLL ↓ | ECE ↓ | cAcc ↑ | cNLL ↓ | cECE ↓ | FLOPs ↓ | Epochs ↓ |
|---|---|---|---|---|---|---|---|---|
| Independent Model* | 96.0 | 0.159 | 0.023 | 76.1 | 1.050 | 0.153 | 3.6e17 | 200 |
| Monte Carlo Dropout* | 95.9 | 0.160 | 0.024 | 68.8 | 1.270 | 0.166 | 1.00x | 200 |
| Snapshot (M=5) | 96.3 | 0.131 | 0.015 | 76.0 | 1.060 | 0.121 | 1.00x | 200 |
| Fast Geometric (M=12) | 96.3 | 0.126 | 0.015 | 75.4 | 1.157 | 0.122 | 1.00x | 200 |
| Prune and Tune (M=6) | 96.5 | 0.113 | 0.005 | 76.2 | 0.972 | 0.081 | 0.85x | 200 |
| Stochastic Annealing (M=6) | **96.7** | **0.110** | **0.005** | **76.3** | **0.968** | **0.079** | **0.85x** | **200** |
| TreeNet (M=3)* | 95.9 | 0.258 | 0.018 | 75.5 | 0.969 | 0.137 | 1.52x | 250 |
| BatchEnsemble (M=4)* | 96.2 | 0.143 | 0.021 | 77.5 | 1.020 | 0.129 | 4.40x | 250 |
| Multi-Input Multi-Output (M=3)* | 96.4 | 0.123 | 0.010 | 76.6 | 0.927 | 0.112 | 4.00x | 250 |
| FreeTickets (EDST) (M=7)* | 96.4 | 0.127 | 0.012 | 76.7 | 0.880 | 0.100 | 0.57x | 850 |
| FreeTickets (DST) (M=3)* | 96.4 | 0.124 | 0.011 | 77.6 | 0.840 | 0.090 | 1.01x | 750 |
| Dense Ensemble (M=4)* | 96.6 | 0.114 | 0.010 | 77.9 | 0.810 | 0.087 | 1.00x | 800 |

| Methods (CIFAR-100/WRN-28-10) | Acc ↑ | NLL ↓ | ECE ↓ | cAcc ↑ | cNLL ↓ | cECE ↓ | FLOPs ↓ | Epochs ↓ |
|---|---|---|---|---|---|---|---|---|
| Independent Model* | 79.8 | 0.875 | 0.086 | 51.4 | 2.700 | 0.239 | 3.6e17 | 200 |
| Monte Carlo Dropout* | 79.6 | 0.830 | 0.050 | 42.6 | 2.900 | 0.202 | 1.00x | 200 |
| Snapshot (M=5) | 82.1 | 0.661 | 0.040 | 52.2 | 2.595 | 0.145 | 1.00x | 200 |
| Fast Geometric (M=12) | 82.3 | 0.653 | 0.038 | 51.7 | 2.638 | 0.137 | 1.00x | 200 |
| Prune and Tune (M=6) | 82.7 | 0.634 | 0.013 | 52.7 | 2.487 | 0.131 | 0.85x | 200 |
| Stochastic Annealing (M=6) | **83.1** | **0.633** | **0.010** | **52.8** | **2.440** | **0.131** | **0.85x** | **200** |
| TreeNet (M=3)* | 80.8 | 0.777 | 0.047 | 53.5 | 2.295 | 0.176 | 1.52x | 250 |
| BatchEnsemble (M=4)* | 81.5 | 0.740 | 0.056 | 54.1 | 2.490 | 0.191 | 4.40x | 250 |
| Multi-Input Multi-Output (M=3)* | 82.0 | 0.690 | 0.022 | 53.7 | 2.284 | 0.129 | 4.00x | 250 |
| FreeTickets (EDST) (M=7)* | 82.6 | 0.653 | 0.036 | 52.7 | 2.410 | 0.170 | 0.57x | 850 |
| FreeTickets (DST) (M=3)* | 82.8 | 0.633 | 0.026 | 54.3 | 2.280 | 0.140 | 1.01x | 750 |
| Dense Ensemble (M=4)* | 82.7 | 0.666 | 0.021 | 54.1 | 2.270 | 0.138 | 1.00x | 800 |

Table 2: Results for ensembles of WideResNet-28-10 models on both CIFAR-10 and CIFAR-100. Methods with * denote results obtained from Havasi et al. (2021); Liu et al. (2022a). Best low-cost ensemble results are **bold**. cAcc, cNLL, and cECE correspond to corrupted test sets. We report the mean values over 10 runs for stochastic annealing.

method. Tables are organized into two groups based on training cost. The first group consists of low-cost training methods that take approximately as long as a single network would take to train. The second group of methods use either significantly more epochs or compute per epoch to achieve comparable performance.

## 6    CONCLUSIONS

Stochastic Subnetwork Annealing offers a novel approach to tuning pruned models by representing subnetworks with probabilistic masks. Rather than discretely removing parameters, we instead create probability matrices that alter the chance for parameters to be retained on any given forward pass. We then anneal those probability values towards a deterministic binary mask over several epochs such that the subnetwork is slowly revealed. We introduce several variations for implementing this idea of subnetwork annealing. Random annealing uses random probabilities for every parameter while temperature annealing applies an even amount of stochasticity to all parameters not in the target subnetwork.

The efficacy of Stochastic Subnetwork Annealing is built upon the same principles behind iterative pruning. Recent insights revealed in optimization research relating to the impact of warmup have revealed that early epochs are critical during optimization as they set the foundational trajectory for the rest of the training process. Gilmer et al. (2021); Ma & Yarats (2021); Gotmare et al. (2018). Stochastic Subnetwork Annealing provides effective regularization during the early epochs of subnetwork tuning to promote training stability and encourage robust adaptation.

Our experiments display marked improvement over the established one-shot and iterative pruning benchmarks for subnetworks at various levels of sparsity. This technique is especially effective for very sparse models up to 98%. We conduct an extensive ablation study to explore the dynamics of this technique with regard to different hyperparameters, pruning methodologies, and learning rate schedules. We additionally implement this technique in the context of Prune and Tune Ensembles where we report significantly better performance against benchmark methods.

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
