# OpenReview forum: "Stochastic Subnetwork Annealing: A Regularization Technique for Fine Tuning Subnetworks"
_ICLR.cc/2024/Conference — Submitted to ICLR 2024_

### Official Review · Reviewer_wViy · 2023-10-31

**Soundness:** 2 fair
**Presentation:** 3 good
**Contribution:** 2 fair
**Rating:** 6
**Confidence:** 3

**Summary:**

This paper introduces an novel tuning approach that employs stochastic masks to represent subnetworks. These masks probabilistically include or exclude pruned parameters on each forward pass during tuning. The method gradually anneals these probabilities over time, which may prevent the subnetworks from getting stuck in lower performing regions of the optimization landscape, thus lead to a smoother optimization of subnetworks at high sparsity levels. Experimental results demonstrate the proposed approach outperforms   one-shot and iterative pruning baselines.

**Strengths:**

- Instead, it presents a novel tuning approach involving retraining pruned neural networks with stochastic masks, in order to help escape suboptimal minima at high sparsities. The proposed tuning method is straightforward and easy to implement.
- This paper also evaluate the efficacy of Stochastic Subnetwork Annealing in the context of the low-cost ensemble algorithm. This provides an interesting perspective on training stochastic subnetworks

**Weaknesses:**

- Several methods exist for retraining pruned neural networks, including fine-tuning with a small fixed learning rate, learning rate rewinding, and weight rewinding (known as the lottery ticket hypothesis) [1,2,3]. It's unclear how the proposed method compares to these alternatives.
- The experimental results primarily rely on small datasets like CIFAR10 and CIFAR100. I am wondering if this approach would yield similar benefits on larger datasets such as ImageNet.
- The use of stochastic masks shares similarities with research on dynamic pruning and training during pruning. However, the paper doesn't discuss this related research line, which could provide valuable context

[1] Renda, Alex, Jonathan Frankle, and Michael Carbin. "Comparing rewinding and fine-tuning in neural network pruning." *arXiv preprint arXiv:2003.02389* (2020).

[2] Le, Duong H., and Binh-Son Hua. "Network pruning that matters: A case study on retraining variants." *arXiv preprint arXiv:2105.03193* (2021).

[3] Frankle, Jonathan, and Michael Carbin. "The lottery ticket hypothesis: Finding sparse, trainable neural networks." *arXiv preprint arXiv:1803.03635* (2018).

**Questions:**

- Retraining techniques like learning rate rewinding and weight rewinding also claim that they prevent the neural network from being trapped in suboptimal regions of the optimization landscape. I'm curious about what advantages the proposed method offers in comparison to these techniques and how it excels in this regard.

---

### Official Review · Reviewer_8ZvZ · 2023-11-06

**Soundness:** 3 good
**Presentation:** 3 good
**Contribution:** 3 good
**Rating:** 5
**Confidence:** 5

**Summary:**

This paper introduces Stochastic Subnetwork Annealing method for optimizing pruned neural networks through probabilistic masks, enhancing subnetwork tuning by allowing a gradual transition from high to low stochasticity. It aims to address the challenges in conventional pruning methods that often result in an initial drop in accuracy, offering a smoother and more robust optimization process. Comparative experiments reveal that this technique outperforms both one-shot and iterative pruning methods, particularly in highly sparse subnetworks. Moreover, the method demonstrates its versatility by improving the performance of subnetwork ensembles by applying the stochastic subnetwork annealing to the Prune and Tune Ensembling strategy, yielding superior performance compared to benchmark methods such as FreeTickets and Dense Ensembling.

**Strengths:**

* The proposed Stochastic Subnetwork Annealing is simple yet effective pruning technique.
* Unlike traditional methods that abruptly remove parameters, this technique uses probability matrices to control parameter retention during forward passes, and the gradual annealing of these matrices ensures a more consistent unveiling of the subnetwork.
* The ensembling strategy also benefits with the proposed approach without a lot of modifications

**Weaknesses:**

* The technique's performance is contingent on the careful selection of annealing functions and the right initialization of temperatures, which might not be straightforward for all users.
* While the approach shows promise in specific contexts, its general applicability across varied architectures and datasets remains to be validated extensively.
* The added computational cost of handling probabilistic masks, especially in large-scale networks, may offset some of the efficiency gains from pruning.
* The idea of stochastic/binary masks has been previously proposed in the literature ([1]), and the ideas such as reverse dropout already exist, so the novelty seems to be lacking.


[1] Kang, Haeyong, et al. "Forget-free continual learning with winning subnetworks." International Conference on Machine Learning. PMLR, 2022.

**Questions:**

* Have you quantified the diversity in the produced ensembles
* Why were the number of ensembles set to 6, does the performance improve or decrease as a function of number of ensembles?
* The baseline approaches for ensembling (e.g., FreeTickets) have adopted the comparison with ResNet50/ImageNet. Was this evaluated for the proposed approach?

---

### Official Review · Reviewer_kiVy · 2023-11-06

**Soundness:** 2 fair
**Presentation:** 2 fair
**Contribution:** 2 fair
**Rating:** 3
**Confidence:** 4

**Summary:**

This paper proposes "Stochastic Subnetwork Annealing", a regularization technique that uses probabilistic masks to include or exclude parameters during each forward pass, rather than removing them outright. To this end, they interpret the pruned mask is determined with a Bernoulli realization of the probability matrix. The author conducts the experiment under CIFAR-10, and CIFAR-100 to show the effectiveness. Furthermore, they show an additional application to network ensemble.

**Strengths:**

The ablation study of the annealing is well conducted.

The method is simple and easy to follow.

**Weaknesses:**

(1) Overall, I think the novelty is somewhat limited. The main contribution of this paper is to use randomness to the binary mask to prune the network, which is highly similar to dropout. Note that there exist papers that utilize dropout for network sparsification [1], and performance improvement due to such randomness is somewhat trivial (as various dropout papers highlighted [2,3])

(2) The overall improvement is marginal. The main baseline is to improve iterative magnitude pruning (IMP), but as shown in Table 1 (the final two tables), the improvement is quite marginal, e.g., 0.564 to 0.566. Furthermore, I think it is not important to improve iterative random pruning or random pruning since IMP is the most important baseline. Finally, it would be great if the author could consider combining other pruning schemes with the proposed method, e.g., [4,5,6].

(3) It would be great to include large-scale datasets as network pruning frameworks sometimes show the inconsistency of trends when scaling to high-resolution images [7]. I think it would be great if the author could show ImageNet, but Tiny-ImageNet or sub-sampled ImageNet would also be great.

(4) For weight pruning experiments, it is good to show the pruned results as follows (examples: Figure 2 of [4] or Table 2 of [8]).
- The initial accuracy and the best accuracy (as it sometimes improves the performance)
- Variance of the result (as pruning, highly depends on the initialization). Especially, as the current improvement is marginal, it would be great to add the variance by running multiple times with different random seeds.

(5) Can the author explain why this method shows better robustness than other ensemble methods? I think there is no component that forces robustness in the suggested method.

(6) More rigorous experiments on various architecture is required.
- The author only considers Wide-ResNet for the main experiments and considers ResNet for the ablation study (which is also a little inconsistent).
- Network pruning should be tested under various architectures to ensure consistency.

(7) minor: Editorial comment.
- The background section consists of various literature. It will be much better to divide the subsection (or use a subsubsection) to group similar backgrounds together.
- Missing citation in Section 5.2 (third paragraph)

[1] Variational Dropout Sparsifies Deep Neural Networks, ICML 2017\
[2] Dropout: A Simple Way to Prevent Neural Networks from Overfitting, JMLR 2014\
[3] Variational Dropout and the Local Reparameterization Trick, NeurIPS 2015\
[4] Layer-adaptive sparsity for the Magnitude-based Pruning, ICLR 2021\
[5] A Signal Propagation Perspective for Pruning Neural Networks at Initialization, ICLR 2020\
[6] Movement Pruning: Adaptive Sparsity by Fine-Tuning, NeurIPS 2020\
[7] Sparse Training via Boosting Pruning Plasticity with Neuroregeneration, NeurIPS 2021\
[8] Training Your Sparse Neural Network Better with Any Mask, ICML 2022

**Questions:**

While I can see the claim of loss landscape in the abstract and the introduction, I cannot find the empirical evidence in other parts. Can the author point out where I can find this support for the claim?

---

### Official Review · Reviewer_i1oq · 2023-11-08

**Soundness:** 2 fair
**Presentation:** 2 fair
**Contribution:** 2 fair
**Rating:** 6
**Confidence:** 3

**Summary:**

The manuscript presents a new pruning approach, named Stochastic Subnetwork Annealing, aimed at optimizing subnetworks within deep neural networks. The proposed method introduces stochastic masks to create a probabilistic framework for parameter inclusion during the training process. This approach seeks to minimize the decline in accuracy typically seen with extensive parameter pruning and to avoid the tendency of models to become too closely fitted to specific, limited areas of the error landscape. The authors report improvements over one-shot and iterative pruning benchmarks on CIFAR-10 and CIFAR-100 datasets.

**Strengths:**

1. The paper combines innovative method, subnetworks, stochastic masks and annealing into the pruning technique.
2. The reported experimental results show a notable improvement, with over a 1% increase in accuracy over established one-shot and iterative pruning benchmarks at various sparsity levels.
3. The authors reported 6% and 4% enhancements in performance over the one-shot pruning baseline at 98% sparsity under constant and one-cycle rate schedules. Initial results demonstrate improvements in model accuracy at various levels of sparsity, which is promising.
4. The authors have conducted a comprehensive comparison of their approach against a broad spectrum of competitive benchmarks
5. The writing is clear.

**Weaknesses:**

1. The experiments are limited to CIFAR-10 and CIFAR-100 datasets. The lack of experimentation on different types of data (e.g., text) questions the generalizability of the proposed method.
2. The authors target at the deep neural network, i.e. ResNet-18 for the testing. Given the significant differences in architecture and scale when compared to more advanced models like transformers, it is uncertain whether the proposed technique would be as effective with these state-of-the-art neural networks. This limitation casts doubt on the applicability of the work.
3. The paper does not compare its results with other stochastic mask pruning methods, which could provide a more comprehensive understanding of the method's relative performance.
4. There is a lack of clarity on the hyperparameters used for comparison with Havasi et al. (2021) and Liu et al. (2022a), which makes it challenging to assess the fairness and validity of the comparisons.
5. The paper does not discuss parallel efficiency, which is crucial for understanding the practical implications of the proposed method in real-world applications.

**Questions:**

1. Can you clarify the hyperparameters used in your comparisons with Havasi et al. (2021) and Liu et al. (2022a)? What does "M" represent within the context of your experiments?
2. Have you conducted experiments on datasets other than CIFAR-10 and CIFAR-100, and if not, do you plan to? And have you tested this pruning framework for more advanced models like transformers?
3. Could you comment on the stochastic Mask pruning method?
    1. Elkerdawy, Sara, et al. "Fire together wire together: A dynamic pruning approach with self-supervised mask prediction." *Proceedings of the IEEE/CVF Conference on Computer Vision and Pattern Recognition*. 2022.
    2. Hayou, S., He, B., & Dziugaite, G. K. (2021). Probabilistic fine-tuning of pruning masks and PAC-Bayes self-bounded learning. *arXiv preprint arXiv:2110.11804*.

---

### Official Review · Reviewer_6mmk · 2023-11-09

**Soundness:** 2 fair
**Presentation:** 4 excellent
**Contribution:** 2 fair
**Rating:** 3
**Confidence:** 3

**Summary:**

The authors propose Stochastic Subnetwork Annealing (SSA), an approach to pruning that replaces traditional pruning masks with matrices of per-parameter probabilities used to sample subnetworks at each iteration. Over time, the probabilities decline for all but a specified percentage of the original parameters. It is claimed that this approach allows for more robust optimization and ultimately better performance.

**Strengths:**

The paper is clearly written (I particularly appreciate the survey-style background section). The method is fairly intuitive and easy to follow. Experiments are fairly exhaustive.

**Weaknesses:**

At a high level, it's unclear to me what this paper accomplishes. It does not offer new insights about pruning, nor does it permit more efficient model inference or training in practice. Accuracy gains over other pruning methods are claimed, but they are fairly marginal, leaving me unconvinced that this is even useful as a proof of concept, in the way that e.g. the original lottery ticket hypothesis paper was.

More specifically:
> However, these models also bring important challenges related to computational needs, storage cost, and training efficiency. The resource-intensive nature of these large networks have spurred a growing interest in techniques that can reduce the size and computational complexity associated with training and deploying these models.

- This sentence from the introduction seems to promise a method that improves the efficiency of practical large models. Like other pruning papers, this one does not provide that. As the authors note themselves, translating unstructured sparsity into efficiency or reduced complexity in practice is very difficult, if not impossible, with current tools. Also, this paper does not evaluate SSA on models at a scale that would benefit from shrinking in practice.

> The probabilistic inclusion of extra parameters early in the tuning process allows for gradient information to bleed through into the target subnetwork, encouraging robust adaptation and avoiding the drastic performance collapse observed with one-shot pruning methods.

> Allowing other parameters to become active allows for gradient information to contribute to optimization of the target subnetwork which can help to encourage avoidance of local minima

- Several claims (^) of this nature are made, but as far as I can tell there is little evidence in the paper to support them. There are no robustness evaluations, as far as I can tell, and avoiding the drastic performance collapse of one-shot pruning is not unique to SSA.

- Table 1 should have full-network accuracy on it. It should also ideally have error bars, since the margins are so slim. Would also be nice to see CIFAR-10 results here, to allow for easier comparison to canonical papers in the field, like the lottery ticket hypothesis paper.

- Table 2 claims that SSA requires fewer FLOPs than some competing methods, but this is not particularly meaningful, in that the method does not actually translate into improved performance on existing ML hardware (right?). Alternatively, one could argue that the burden of having to keep around float matrices of probabilities the same size as the network also outweighs theoretical improvements to FLOP counts, since practical models most in need of being pruned are now tens or even hundreds of gigabytes.

- The differences between SSA and regular Prune and Tune in Table 2 seem far too close to call for me. I'd need to see error bars here as well.

Bits and bobs (no bearing on decision):
- There's a missing citation (marked by a ?) on page 8.

**Questions:**

- I'm a little confused by the fact that iterative pruning is so close to one-shot pruning in most of the columns of Table 1. I was under the impression that iterative pruning was supposed to be substantially better. Do the authors have a sense of why this is?

---

### Author Response · Authors · 2023-11-19

I want to thank the reviewers for their time and their feedback. In response to those that recommended the decision to reject, I wish to clarify and counter several points that were raised.

1. Novelty of Methodology: Dropout, sparsity inducing regularization, network gating, and other pruning techniques suggested by the reviewers have completely distinct methodologies and purposes to the work we introduce in this paper. Our approach focuses on the optimization of any pruned subnetwork from a pre-trained model in a small number of epochs. This is a significant departure from the suggested methods, which involve routines that learn a single, specific subnetwork throughout the entirety of training. Our use of stochastic annealing is a novel concept in this context, and we believe it reveals valuable insights relating to regularization in the early phases of fine tuning.

2. Significance of Improvement: Our work demonstrates substantial improvement for optimizing random subnetworks over other established methods. This improvement holds significant value for a wide variety of pruning techniques used in practice, especially those that leverage multiple subnetworks from a single model. The Prune and Tune Ensemble experiment underscores this value. We also demonstrate consistent improvement for tuning subnetworks pruned by weight magnitude. The smaller margins are natural as these subnetworks are more optimized than random subnetworks. However, this does not invalidate the value of our approach which still reliably enhances their effectiveness at various levels of sparsity, and this notion of smaller margins can be observed at the top of many competitive leaderboards. Some of the largest breakthroughs in ML were built on meaningful insights derived from papers with incremental improvements.

3. Computational Efficiency: There seems to be a misunderstanding about the computational efficiency of our approach. Our method anneals the float matrices towards a deterministic binary structure. These float matrices can be tossed out after tuning, resulting in small and sparse subnetworks like you would get from any other pruning method. Our method is fully compatible with any arbitrary pruning structure to allow for easier hardware optimization.

4. More Experiments: The models and baselines we used are well established in the literature. Models of this scale are widely used in practice, especially in resource constrained environments. While Transformers are growing in popularity, Resnets are still the most popular network architecture in computer vision, and recent research has shown that CNNs are as performant as Transformers at scale [4]. Our robustness claims are evaluated in the context of the ensemble experiment via uncertainty baselines [1]. Suggestions for further experimentation on different modalities, while valuable, may be more suited to a comprehensive study beyond the scope of this paper.

Error was not included in the tables on the first draft due to layout constraints. We instead reported accuracy to the tenths place, which is common in recent published ensemble literature [1,2,3]. However, in response to the reviewers, we reworked the table to include standard error over 3 more additional runs and added a note on the parent network's accuracy to Table 1.

Despite the concerns, there is consensus among the reviewers for our clear writing, intuitive methodology, and exhaustive experiments. We believe this further reinforces the potential value and impact that our work holds and we hope this merits consideration for acceptance.

[1]: Google Uncertainty Baselines, https://github.com/google/uncertainty-baselines

[2]: Marton Havasi, Rodolphe Jenatton, Stanislav Fort, Jeremiah Zhe Liu, Jasper Snoek, Balaji Laksh-minarayanan, Andrew M. Dai, and Dustin Tran. Training independent subnetworks for robust prediction, 2021.

[3]: Shiwei Liu, Tianlong Chen, Zahra Atashgahi, Xiaohan Chen, Ghada Sokar, Elena Mocanu, Mykola Pechenizkiy, Zhangyang Wang, and Decebal Constantin Mocanu. Deep ensembling with no overhead for either training or testing: The all-round blessings of dynamic sparsity, 2022a.

[4]: Samuel L. Smith, Andrew Brock, Leonard Berrada, Soham De, ConvNets Match Vision Transformers at Scale, https://arxiv.org/abs/2310.16764

---

### Meta-Review · Area_Chair_25vB · 2023-12-06

**Metareview:**

The authors introduce a probabilistic neural network pruning method motivated by the argument that discrete removal operations can lead to overfitting in local regions of the loss landscape. The method anneals the probability of parameters being included in any given forward pass over time to a pre-determined target sparsity level. In addition, an ensembling strategy extension is being presented.

Reviewers found the manuscript to be clearly written and easy to follow and the method to be intuitive and easy to understand. However, the method was described as lacking novelty (Reviewers kiVy, 8ZvZ) and leading to only marginal improvements (Reviewers 6mmk, kiVy). Experimental evaluation was found to be limited (Reviewers i1oq, kiVy, wViy) and several other issues were mentioned (unclear effect on efficiency, robustness arguments not convincing).

**Justification For Why Not Higher Score:**

My personal view is in agreement with the reviewers' arguments, and while I read the authors' responses, I was not convinced by the counterarguments offered. The fact that no concrete effort was made to improve the manuscript (e.g. include additional experiments) after the first round of reviews is a further justification for rejection.

**Justification For Why Not Lower Score:**

N/A

---

### Decision · Program_Chairs · 2024-01-16

Reject